# Improved Bioavailability of Montelukast through a Novel Oral Mucoadhesive Film in Humans and Mice

**DOI:** 10.3390/pharmaceutics13010012

**Published:** 2020-12-23

**Authors:** Johanna Michael, Diana Bessa de Sousa, Justin Conway, Erick Gonzalez-Labrada, Rodolphe Obeid, Julia Tevini, Thomas Felder, Birgit Hutter-Paier, Horst Zerbe, Nadine Paiement, Ludwig Aigner

**Affiliations:** 1Institute of Molecular Regenerative Medicine, Spinal Cord Injury and Tissue Regeneration Center Salzburg, Paracelsus Medical University, 5020 Salzburg, Austria; johanna.michael@pmu.ac.at (J.M.); diana.bessa@pmu.ac.at (D.B.d.S.); 2IntelgenX Corp., Saint-Laurent, QC H4S 1Y2, Canada; justin@intelgenx.com (J.C.); erick@intelgenx.com (E.G.-L.); rodolphe@intelgenx.com (R.O.); horst@intelgenx.com (H.Z.); 3Department of Laboratory Medicine, Paracelsus Medical University, 5020 Salzburg, Austria; j.tevini@salk.at (J.T.); t.felder@salk.at (T.F.); 4QPS Neuropharmacology, 8074 Grambach/Graz, Austria; birgit.hutter-paier@qps.com; 5Austrian Cluster of Tissue Regeneration Affiliation, 1200 Vienna, Austria

**Keywords:** montelukast, drug delivery, Alzheimer’s disease, dementia

## Abstract

The leukotriene receptor antagonist Montelukast (MTK) is an approved medication for the treatment of asthma and allergic rhinitis. The existing marketed tablet forms of MTK exhibit inconsistent uptake and bioavailability, which partially explains the presence of a significant proportion of MTK low- and non-responders in the population. Besides that, tablets are suboptimal formulations for patients suffering from dysphagia, for example, seen in patients with neurodegenerative diseases such as Alzheimer’s disease, a disease with increasing interest in repurposing of MTK. This, and the need for an improved bioavailability, triggered us to reformulate MTK. Our aim was to develop a mucoadhesive MTK film with good safety and improved pharmacological features, i.e., an improved bioavailability profile in humans as well as in a mouse model of Alzheimer’s disease. We tested dissolution of the MTK mucoadhesive film and assessed pharmacoexposure and kinetics after acute and chronic oral application in mice. Furthermore, we performed a Phase I analysis in humans, which included a comparison with the marketed tablet form as well as a quantitative analysis of the MTK levels in the cerebrospinal fluid. The novel MTK film demonstrated significantly improved bioavailability compared to the marketed tablet in the clinical Phase 1a study. Furthermore, there were measurable amounts of MTK present in the cerebrospinal fluid (CSF). In mice, MTK was detected in serum and CSF after acute and chronic exposure in a dose-dependent manner. The mucoadhesive film of MTK represents a promising alternative for the tablet delivery. The oral film might lower the non-responder rate in patients with asthma and might be an interesting product for repurposing of MTK in other diseases. As we demonstrate Blood-Brain-Barrier (BBB) penetrance in a preclinical model, as well as in a clinical study, the oral film of MTK might find its use as a therapeutic for acute and chronic neurodegenerative diseases such as dementias and stroke.

## 1. Introduction

Montelukast (MTK) is a leukotriene receptor antagonist commonly used to treat patients suffering from chronic asthma as well as seasonal allergies and allergic rhinitis. MTK binds to the cysteinyl-leukotriene receptor 1 and to the leukotriene receptor GPR17 with high affinity and selectivity, thereby blocking the leukotriene signaling pathway [1,2]. It reduces leukotriene-mediated respiratory inflammation reactions, such as vasoconstriction, and relieves asthma symptoms [3]. Currently, MTK is marketed under the brand name Singulair^®^ and in several generic products in the oral tablet and other pharmaceutical forms, such as oral granules. The available forms present a number of limitations such as inconsistent solubility, uptake, and bioavailability, for several reasons. Although MTK is freely soluble in water, its solubility increases significantly above pH 7.5 and drastically reduces under acidic conditions normally found in the gastrointestinal tract—in particular, in the stomach [4]. This has led to relatively slow and inconsistent absorption into the blood stream, with maximum concentrations occurring between 2 and 4 h following consumption, thereby limiting its use to chronic applications rather than for rapid acute treatment. The uptake and bioavailability of MTK are further determined by pharmacogenetics (for review, see [5]). For example, more than 20% of the population is not responding to MTK with a clinical benefit [6]. Among the various genetic reasons are variations in the SLCO2B1 gene coding for the organic anion transporting OATP2B1, which has been associated with altered absorption of MTK [7]. Uptake of MTK was further modified by the intake of citrus juice [8]. Besides the physico-chemical and genetic basis for the insufficient uptake and bioavailability of MTK in its current tablet form, a further drawback is the general inadequateness of the tablets for patients suffering from dysphagia, such as elderly patients, patients with dementia, and patients that require intubation or ventilation. In summary, the tablet form of MTK represents a number of limitations, and, consequently, there is an increasing interest in developing improved strategies for the delivery of MTK, aiming for increased bioavailability [9].

There is already a certain rationale for MTK to be repurposed for Alzheimer’s disease (AD) (for review, see [10]). Leukotriene production is elevated in the AD brain [11,12]; leukotrienes are involved in various aspects of AD pathologies, in particular, neuroinflammation and neuronal cell death, and genetic and pharmacological blockage of leukotrienes in animal models of neurodegenerative diseases, including AD, have demonstrated efficacy in reducing pathology and promoting cognitive functions (reviewed in [10]). Finally, MTK is associated with a reduced risk of dementia, and a recent case report suggested that MTK might improve cognitive function in patients with dementia [13,14,15]. Oral mucoadhesive application has several advantages, such as the relatively easy and convenient accessibility of the oral cavity, the limiting of first-pass effects and degradation during the gastro-intestinal transit and the rapid drug absorption due to high blood circulation in the mucosa (as reviewed in [16,17]). In general, buccal films are seen as a promising alternative application route for drugs as they combine several features, such as better patient compliance (no swallowing and the possibility to modify the taste) and increased bioavailability (reviewed in [18]).

We developed a novel MTK oral mucoadhesive film to circumvent the limitations of MTK in its tablet form, tested its stability and dissolution, and performed a Phase I bioavailability and safety study in healthy volunteers, where we assessed MTK pharmacokinetics in serum and cerebrospinal fluid. Furthermore, we wanted to investigate the possibility of repurposing MTK in the form of the novel oral film for Alzheimer’s disease (AD). To test its suitability for a pre-clinical efficacy study of MTK in AD, we assessed pharmacokinetics and tested the effects of acute and chronic application on pharmacoexposure in an animal model of AD, the 5xFAD mouse model.

## 2. Materials and Methods

### 2.1. Film Preparation

The mucoadhesive films were produced by a solvent-casting technique. For that, excipients and active pharmacological ingredient (API) in a range of 0–30.00% (*w/w*) were mixed and dissolved/suspended in water. The resulting wet blend was spread to a thickness less than 1 mm onto a release liner. The wet film was dried in an oven for 60–75 min at 65 °C. The dried film sheet was cut into film strips containing 10 mg of MTK. The composition of the film is depicted in Table 1.

### 2.2. In Vitro Dissolution Test

Dissolution studies were carried out in a paddle type apparatus at 50 rpm and 37 °C. Dissolution of MTK tablet, MTK mucoadhesive film and pre-solubilized MTK oral film was assessed in 900 mL of phosphate based saliva buffer (pH 6.8). For the dissolution test of pre-solubilized MTK film, a single film unit was mixed with 2 mL of saliva buffer pH 6.8 for 5 min and directly injected into the dissolution chamber. Aliquots of the release medium (8 mL) were collected at the following timepoints: 2.5, 5, 7.5, 10, 15, 20, 30, 45 min and analyzed by HPLC UV at 255 nm as specify in the USP (United States Pharmacopiea).

### 2.3. Mechanical Properties

To determine potential impact of cold temperature during transport films containing 10 mg MTK were placed at −20 °C for 15 and 30 days, respectively, and were compared to a control product unexposed to cold temperature. The mechanical properties were evaluated by testing 3 films for folding endurance, and 8 films for elongation and tensile strength. A Mecmesin Multitest 1-d force testing system (SN 09-1004-11) (Slinfold, West Sussex, UK), including a Mecmesin Advanced Force Gauge 50N (SN 09-0071-11), was used to perform the tests. The flexibility of a film was assessed by repeatedly folding the film at the same place until cracking or breakage. The number of times the film was folded without breaking was recorded as the folding endurance value. A film strip was fixed at half-length on the bottom grip of the instrument, and then folded 10 times in both directions.

Elongation at break was measured by stretching the film to its maximum deformation until it tore apart. The test was carried out by affixing a film by its ends to the grips of a Mecmesin Multitest 1-d instrument and stretching it at a constant speed of 10 mm/min until it cracked or broke. The length of the film section was measured before and after stretching. Elongation is defined as the ratio between the increase in film length, as result of stretching, and the initial length of the film, expressed as a percentage of the initial length. Tensile strength was determined simultaneously with the elongation test and is defined as the maximum force applied to the film area being stretched. All tests were performed in a room with controlled temperature and humidity.

### 2.4. Clinical Study—Subjects and Study Design

To determine the bioequivalence of MTK, 10 mg mucoadhesive film (test drug—treatment A) and Singulair^®^ 10 mg film-coated tablet (Merck Sharp & Dohme, (Kenilworth, NJ, USA)) (reference drug—treatment B) a single-dose, randomized, open-label, two-way crossover comparative bioavailability pilot study was conducted in volunteers under fasting conditions (ethical approval number MOD00167465). The clinic study was performed by BioPharma Services Inc (BPSI) (Toronto, ON, Canada) and included 8 healthy non-smokers (for at least 6 months prior to first drug administration) male and non-pregnant female volunteers, with 18 years of age or older and a body mass index (BMI) within the range of 18.5–29.9 kg/m^2^. To assess the clinical health of the participants, several laboratory tests were performed. Screenings included demographic data, medical and medication histories, physical examinations, body measurements, vital signs (seated blood pressure (BP), heart rate (HR), respiratory rate (RR) and temperature), electrocardiogram (ECG), hematology, biochemistry, serology, urinalysis, urine screening for drugs of abuse and cotinine, alcohol test and serum pregnancy test (for the full list of clinical laboratory assessment, see Appendix A). Clinical laboratory values needed to be within BPSI’s acceptable range. A single dose of 10 mg MTK of either formulation was administered under fasting conditions. Blood sampling (4 mL) was performed 0.5, 1, 1.5, 2, 2.25, 2.5, 2.75, 3.25, 3.5, 3.75, 4, 4.25, 4.5, 5, 5.5, 6, 8, 12 and 24 h after dosing using a K2EDTA Vacutainer^®^. Additionally, cerebrospinal fluid (CSF) samples were taken 3.0 and 7.0 h post-dose in study period 1. Participants were housed at least 10 h prior to dosing and until 12 h post-dose. The confinement period lasted in total at least 22 h for each study period, with occurrence of adverse effects being closed monitored during this period. An investigator was present from approximately 30 min prior to dosing and until at least 12 h (study period 1) or 4 h (study period 2) after the last subject was dosed. The investigator remained on-call throughout the duration of the study. For safety, in each study period, vital signs (blood pressure (BP) and heart rate (HR)) were monitored before dosing and 3 and 24 h post-dosing. The study lasted for at least 2 weeks from the study period 1 check-in, compromising a washout interval of minimum 7 days between the two clinical phases.

### 2.5. Animals

Acute exposure experiment: Two-month-old C57BL6 female mice (*n* = 10) were treated with a single dose of MTK mucoadhesive film. One group (*n* = 5) received a dose of 3.3 mg/kg/d of MTK, and another group (*n* = 5) received 10 mg/kg/d of MTK. The respective films were punched out of an initial film into circular films using punch pliers with 3 mm diameter, which results in an area of 7 mm^2^ per film. Based on the area of the film punch, we calculated the different concentrations of MTK in the initial film to reach 3.3 mg/kg/d and 10 mg/kg/d, respectively, in a 25 g mouse with one 7mm^2^ film. For oral treatment, mice were held in the mouse grip and the film was placed on the oral mucosa using riffled forceps with a 0.5 diameter at the tip.

To assess serum levels of MTK, blood samples were collected one, three and seven hours after the beginning of treatment. For the first two timepoints, blood was taken from the medial saphenous vein and for the last one by cardiac puncture. Mice were housed at the Paracelsus Medical University Salzburg in groups under standard conditions at a temperature of 22 °C and a 12 h light/dark cycle with ad libitum access to standard food and water. This study was approved by local Austrian ethical authorities (BMBWF-66.019/0019-V/3b/2018).

Chronic exposure experiment: Transgenic 5xFAD mice [19], which have three mutations in the gene for APP695 (APP K670N/M671L (Swedish), I716V (Florida), V717I (London)) and two mutations in the gene for PS1 (M146L, L286V) under the expression of the Thy1 promotor, were used for this experiment. Mice were housed at the animal facility of QPS Austria in groups under standard conditions with constant 12 h light/dark cycle, at a room temperature of 24 °C, relative humidity of 40–70%, and ad libitum access to standard rodent chow (altrumin) and water. The experiment was approved by the local ethical committee (ABT13-14688/2018-4) and conducted at QPS. In this study, 5-month-old animals (*n* = 44) were treated daily with MTK mucoadhesive film for 13 weeks. The film was placed on the mice buccal mucosa, and two different doses were used—3.3 mg/kg/d MTK (referred to as low dose) or 10 mg/kg/d MTK (referred to as high dose). As a control, animals received vehicle treatment (film without API). Body weights were recorded by weighing all of the mice every week on day one of the week using a digital scale. For the assessment of fecal motility, the number of fecal boli were counted after the mice spent a period of 20 min in an open field. The number of boli per mouse was used as the read-out parameter for fecal motility.

### 2.6. Quantitative Assessment of MTK in Plasma and CSF Human Samples

Serum and CSF were prepared as described in our previous work [20]. Samples were analyzed using an Agilent 1290 Infinity binary pump and autosampler with detection by an Agilent 6550 iFunnel QTof mass spectrometer (MS) (Agilent) using electrospray ionization. The protonated molecules for MTK and tolbutamide (m/z 586.2177 and m/z 271.1116) were extracted with ±15 ppm mass windows to generate chromatograms with suitable combinations of specificity and signal/noise. Sample aliquots (5 µL) were injected into a mobile phase initially of 98% water/0.1% formic acid (Channel A) and 2% acetonitrile/0.1% formic acid (Channel B) delivered at 0.4 mL/min to an Acquity BEH C18 50 × 2.1 mm, 1.7 µm column. The column was maintained at 50 °C in an Agilent Infinity column oven. The mobile-phase composition was programmed to change linearly from 2% Channel B at 0.30 min post-injection up to 95% at 1.10 min, maintained at 95% until 1.75 min, and then returned to initial composition at 1.8 min. Column effluent was diverted to waste for the first 0.8 min post-injection to minimize source contamination. Data processing: MassHunter software (v B.05.01, Agilent UK). Calibration curves were fitted using the simplest regression model to minimize back-calculated calibration standard concentration residuals over the range of the study sample concentrations.

### 2.7. Quantitative Assessment of MTK in Serum and CSF Mouse Samples (HPLC-MS/MS)

The LC–MS/MS method for the quantification of MTK in serum and CSF was described earlier [21] with a modified sample preparation as used in [22]. Briefly, sample preparation consisted of a simple protein precipitation protocol. Therefore, 12.5 μL 100% FA was added to 50 μL of serum or CSF, vortexed briefly before the addition of 150 μL 100% ACN containing MTK-d6 as internal standard. After vortexing for two minutes, all samples were centrifuged at 10,500× *g* for 10 min at 4 °C. For serum samples, 20 μL of the clear supernatant was added to 40 μL of mobile phase A (1 mM ammonium formate in water containing 0.1% FA). The supernatants (180 μL) of CSF extraction were dried under a constant nitrogen flow at 45 °C. Completely dried samples were reconstituted with 60 μL 1 mM ammonium formate in 25/75 (vol/vol) acetonitrile/water containing 0.1% FA. Chromatographic separation was carried out on an Agilent 1200 series quaternary HPLC system using a Chromolith Performance RP18-e column (100 × 3 mm) from Merck, operated at a temperature of 45 °C, with 1 mM ammonium formate in water containing 0.1% FA as mobile phase A, and 1 mM ammonium formate in 95/5 (vol/vol) acetonitrile/water containing 0.1% FA as mobile phase B. Gradient elution at a flow rate of 0.5 mL/min started from 25.0% to 95.0% B in 10.0 min, followed by a flushing step with 95.0% B for 0.8 min followed by an re-equilibration step with 25.0% B for 3.2 min. Total time for a single chromatographic run was 14.0 min. Injection volumes of 20 μL for serum and 30 μL for CSF samples were chosen. Selected reaction monitoring (SRM) measurements for MTK as well as for the d6-internal standard in obtained samples were performed on an API 4000 LC–MS/MS triple quadrupole system in positive ionization mode. The quantifier ion transitions of MS/MS detection were m/z 586.2 to 568.2 for MTK and m/z 592.2 to 574.4 for MTK-d6. Calibration curves were derived from ratios of the peak areas of MTK and the internal standard using 1/χ-weighted linear least-squares regression of the area ratio versus the concentration of the corresponding internal standard MTK-d6. Analyst software 1.6.2 was used for detection, analysis and quantification of data.

### 2.8. Statistical Analysis

For statistical analysis of the human data, analysis of variance (ANOVA) was used for ln-transformed AUCt, AUCinf and C_max_, and untransformed T_max_, λ and T1/2. T_max_ was also analyzed using an additional nonparametric test (Wilcoxon test). The 90% confidence intervals (CIs) for the Test/Reference ratios of geometric means for AUCt, AUCinf and C_max_ were calculated based on the least square means (LSMEANS) and ESTIMATE of the ANOVA.

For statistical analysis of the animal data, Prism 8 software (GraphPad) was used. The data were tested for normal distribution with the Kolmogorov–Smirnov test. For comparison of two groups, an unpaired t test was performed, whereas for comparison of more than two groups, one-way analysis of variance (ANOVA) was used with Tukey’s or Bonferroni’s multiple comparison test as a post-hoc test. The data were depicted as mean and standard error of the mean (SEM) or standard deviation (SD) with a 95% confidence interval as indicated in the respective figure legends. *p* values of *p* < 0.0001 and *p* < 0.001 were considered extremely significant (**** or ***), *p* < 0.01 very significant (**) and *p* < 0.05 significant (*).

## 3. Results

### 3.1. Composition, Preparation, Physical and Mechanical Properties of the MTK Mucoadhesive Film

In addition to the active pharmacological ingredient, the general composition of the film blend includes 60.0–85.0% *w/w* wet of water as solvent and 5.0–15.0% *w/w* wet mucoadhesive polymer to form the film. Furthermore, it includes gum to modify viscosity, stabilizers to prevent degradation, plasticizers to tune mechanical properties, permeation enhancers to increase absorption and colorant and flavor for patient compliance (Table 1). The film was prepared using a solvent-casting technique, where excipients and active pharmacological ingredient (API) were mixed, dissolved in water and spread out to a thin layer that was dried and then cut into strips with a concentration of 10 mg of MTK per strip (Figure 1A).

The 10 mg film has demonstrated stability over 2 years under ICH (International Conference on Harmonization) long-term stability condition (25 °C/60% relative humidity (RH)) and 6 months under ICH accelerated conditions (40 °C/75% RH) without any impurities out of specifications. To assess the impact of exposure to freezing conditions on the mechanical properties of 10 mg MTK film, films were tested for folding endurance, elongation and tensile strength at T_0_ as well as after 15 and 30 days at −20°C (Table 2). Results of folding endurance testing showed that films from all three conditions can be folded at least 10 times without tearing and therefore meet the acceptance criteria of the folding test. A one-way ANOVA analysis was performed to compare the elongation mean values of the three conditions. At a α = 0.05 significance level, the elongation data from films exposed to freezing conditions are not significantly different to the initial product (T_0_). A similar statistical analysis was applied to the tensile strength data and revealed no significant differences between the three stability conditions either.

These results show that the 10 mg MTK film has stable mechanical properties under freezing conditions for at least 30 days. Therefore, the integrity and quality of the film product were maintained in case of exposure to low temperatures during air transport of the films from the manufacturing site to the location where the clinical study took place. 

### 3.2. Dissolution

Release of MTK from films was compared to Singulair^®^ tablets using a USP grade dissolution apparatus (Figure 1B). In general, the dissolution experiments were conducted using a 10 mg dosage unit of either film or tablet. In these experiments, MTK release from coated tablets is compared to that from the mucoadhesive film and pre-solubilized film under pH conditions where full MTK release was observed. The use of a “pre-solubilized” form of film simulated the condition in which the film, after being applied in the oral mucosa, starts to slowly dissolve in saliva and is swallowed.

With MTK film, 80% of the API was released after approximately 6 min, while with the tablets, an equivalent release was only achieved after 10 min. This highlights the rapid oral disintegration advantage of the film-based platform. However, the most significant improvement using the film technology is observed when comparing the tablet to the pre-solubilized MTK film. This experiment is particularly informative as it is a direct comparison of how API is released from swallowed MTK tablets versus swallowed dissolved oral films under the same environmental conditions. This is important, as we cannot exclude that some patients, instead of placing the film on the mouth mucosa, might swallow the film. As a result, the pre-solubilized MTK film reaches 80% released API in only approximately 1 min, whereas the tablet reaches 80% released API after 10 min. This clearly demonstrates how the orally pre-solubilized MTK-film platform releases API more quickly than the MTK-tablet dosage in a neutral pH environment.

### 3.3. Improved MTK Bioavailability in Human Plasma and CSF in MTK Mucoadhesive Film

We have performed a Phase 1 clinical study in healthy human subjects to determine the pharmacokinetics of MTK administered in the form of an oral film product compared to the marketed reference product Singulair^®^, which both contained 10 mg MTK free base (Figure 2A). Enrolled subjects had a mean age of 44 years and a mean BMI of 26 kg/m^2^ (Table 3). Pharmacokinetic parameters of MTK in plasma were collected for each subject individually and are summarized in Table 4.

The area under the concentration–time curve (AUCt), measured from timepoint zero until the last sampling timepoint at 24 h, was significantly higher after the film (AUCt = 3673 ng × h/mL) than after the tablet (AUCt = 2409 ng × h/mL). In this study, the data from the parameters of AUC and C_max_ do not follow a normal distribution, but a lognormal distribution. Therefore, as the use of the geometric mean over the arithmetic mean is favored, here, the calculated film to tablet geometric mean ratio was 152.46%, demonstrating a significantly higher extent of absorption after the film compared to the tablet. Similar results were obtained for the AUCinf, measured from timepoint zero to infinity, calculated as AUCt + C_last_/λ, where C_last_ is the last measurable concentration of MTK. The ratio of film to tablet was 153.15%, again demonstrating a significant higher extent of absorption of the film. Mean pharmacokinetic plasma profiles of MTK for both products over time are shown in Figure 2. The geometric mean of C_max_ was 338 ng after treatment with Singulair^®^ and 554 ng/mL after treatment with the film (Figure 2B). T_max_ was reached at 3.63 h after Singulair^®^ and at 2.63 h after the film (Figure 2B). T_max_ was calculated based on the time at which the maximum concentration is achieved for each individual and not based on the average obtained from the graphical representation. Singulair^®^ demonstrated a plateau-like maximum concentration where T_max_ varied from one subject to the other. The inter-individual variation of T_max_ was less in the MTK mucoadhesive film resulting in a peak at 2.7 h. These results indicate that the MTK film has approximately 1.5 times the C_max_ and AUC (area under the curve) values compared to the Singulair^®^ reference. This indicates that using equivalent API loading, the MTK mucoadhesive film exhibits significantly improved bioavailability. Furthermore, treatment with the film had a one-hour earlier T_max_ than the tablet.

Leukotriene blockers (i.e., leukotriene receptor antagonists and leukotriene synthesis inhibitors) can function to improve cognitive impairment by reducing the neuroinflammatory response within the brain (reviewed in [10,23,24]). Leukotriene blockers, such as MTK, must therefore cross the blood–brain barrier and accumulate in the CSF. To investigate the ability of MTK to cross the blood–brain barrier in humans, CSF levels of MTK were measured 3 and 7 h after dosing with the MTK mucoadhesive film. Measurable amounts, ranging from 3.2 to 4.7 ng/mL of MTK in the CSF, were detected at both timepoints (Table 5).

A comparison of the mean concentrations of MTK in the CSF three hours (3.6 ng/mL) and seven hours (4.2 ng/mL) after application reveals a slower and different pharmacokinetic profile in the CSF than in the serum, where C_max_ was reached after approximately 2.7 h after application. Compared to serum levels the concentration of MTK in the CSF is lower but could be more stable over time. As the mean concentration is slightly higher after seven hours than after three hours, we might not have reached C_max_ after seven hours. However, further studies with longer periods of observation are needed to monitor the temporal profile of MTK pharmacokinetics in the CSF in more detail.

### 3.4. Clinical Safety

Six mild adverse effects (AEs), including somnolence, headache, pruritus, mechanical urticaria and back pain, were experienced by the subjects after taking MTK mucoadhesive film (Table 6), four of which (pruritus, mechanical urticaria, headache and backpain) were classified as unrelated or unlikely related to drug (Table 7). The mild AEs that were classified as possibly related to the drug were somnolence reported by two subjects. No AEs were reported by the subjects after taking the reference product (Table 6). No serious adverse events were reported during the conduct of this study (Table 7).

### 3.5. Pharmacoexposure and Kinetics of MTK in WT and in 5xFAD Mice

As outlined in the introduction, there is a current interest in repurposing of MTK for the treatment of patients with neurodegenerative diseases such as dementia patients [10,13,14,15], which show dysphagia, and, therefore, a mucoadhesive film might have certain advantages. Especially for repurposing of MTK in neurodegenerative diseases, preclinical evidence of efficacy in rodent models is typically required. Therefore, we tested the MTK buccal mucoadhesive film (Figure 3A) in the context of the present study for its pharmacoexposure and kinetic profile in mice, healthy wildtype (WT) mice and 5xFAD mice, a mouse model of genetic AD.

We first briefly assessed the serum pharmacokinetics of the buccal MTK film in 2 months old female C57bl6 wildtype (WT) mice. In a short-term pharmacokinetic experiment, WT animals received one dose of film (Figure 3B) with either 3.3 mg/kg/d (referred to as low dose) or 10 mg/kg/d (referred to as high dose) of MTK (Figure 4A). Regardless of the dose, the highest MTK concentration in serum was detected 1 h post-dosing, whereas measurements 3 and 7 h after the film application showed decreased MTK serum concentrations with time (Figure 4A). The animals receiving the high dose treatment showed, in general, higher MTK serum concentrations, although we observed a high inter-individual variation in serum concentrations (Figure 4A). This suggests variation in the rate of absorption between individuals. Next, we analyzed MTK serum concentration in 5 months old 5xFAD animals after chronic exposure. For that, animals were treated daily for 13 weeks with vehicle, low or high dose of MTK (Figure 3C), and serum MTK levels were assessed 7 h after the last dose (Figure 4B). MTK serum levels were dose-dependent, with animals in the higher dose treatment group showing significantly higher serum concentrations compared to the animals in the lower dose treatment group (low dose: 103.7 ± 15.51 ng/mL, high dose 360.9± 67.5 ng/mL) (Figure 4B). As expected, in the vehicle treated group, no MTK was detected in the serum (Figure 4B). Interestingly, MTK serum levels after chronic exposure were significantly higher in comparison with the serum concentrations determined 7 h after a single dose administration (acute pharmacoexposure) (Figure 4C).

As a proof of concept, we also analyzed MTK concentration in the CSF after chronic exposure and detected higher levels in the high-dose group than in the low-dose group and no MTK in the vehicle group (Figure 4D), demonstrating again the ability of MTK to cross the BBB.

### 3.6. Safety and General Health in Mice

General health status of the animals was monitored during the chronic exposure study by weekly body weight assessment. Mean body weight was stable through the experiment (Figure 5A), with individual body weight fluctuating less than approximately 10% of the initial body weight from week to week (Figure 5B). Mean body weight did not significantly differ between treatment groups (Figure 5C). Furthermore, the fecal motility of the animals was monitored for 20 min. There was no significant difference in the number of fecal boli between the groups (Figure 5D) showing no effect of the new formulation of MTK on fecal motility in 5xFAD mice. In summary, we conclude that prolonged treatment with low and high doses of MTK did not negatively affect general health of mice and no adverse events have occurred in mice.

## 4. Discussion

Montelukast is an approved anti-allergic and anti-asthmatic drug that is commercialized only as solid dosage forms, in particular, as tablets. The limited number of MTK dosage forms available is associated with its physico-chemical properties and sensitivity to light, humidity, temperature and oxidation [9]. The commercialization of MTK in tablet forms might compromise patient adhesion to treatment among groups with swallowing difficulties such as elderly and children. Oral mucoadhesive films circumvent some of the limitations of solid tablets, such as swallowing problems, and could be a promising alternative for MTK treatment with an improved patient compliance. In the present study, we used, for the first time, a novel formulation of MTK, i.e., a mucoadhesive film, designed to be applied in the oral mucosa, in humans and mice.

First, we compared the bioavailability and pharmacokinetic profile of MTK mucoadhesive relative to the reference formulation, Singulair^®^ tablets. For equivalent drug loadings (10 mg), MTK mucoadhesive film exhibits 50% better bioavailability compared to MTK tablets. The superior bioavailability of the MTK mucoadhesive film might be related to (1) sublingual MTK absorption, and/or (2) improved drug release. When applied in the oral mucosa, MTK mucoadhesive film releases the drug into the oral cavity, where it can be absorbed sublingually. By favoring pre-gastric absorption, MTK mucoadhesive film might limit first pass effects resulting in an increased bioavailability [9]. Furthermore, as we showed in vitro, the mucoadhesive film releases MTK almost twice as fast as the tablets, which could as well contribute to an increased MTK bioavailability. Alongside its potential to improve patient compliance, this new formulation opens possibilities for the development of films containing a lower drug-loading bioequivalent to the current reference formulation, Singulair^®^ tablets. Additionally, as MTK mucoadhesive film shows earlier T_max_ and greater C_max_ compared to tablets, it might receive new indications, such as the treatment of acute allergic symptoms, in which a quick drug release and onset of action are desired. In the US, there are some other commercial products that also use a film as drug delivery system. For example, within the course of treating opioid addiction, a sublingual film of Suboxone^®^, containing buprenorphine and naloxone, can be used. Other examples would be Belbuca^®^, a buccal film containing buprenorphine used for pain treatment and Kynmobi, apomorphine hydrochloride, a sublingual treatment used in Parkinson’s disease. However, the number of products on the market is very limited.

As evidence arises from animal experiments and from case reports that MTK might be a valuable therapeutic in the field of neurodegenerative diseases [13,14,15,23,24], we found it relevant to investigate whether MTK can be detected in human CSF, as a proof of concept that MTK can pass the blood–brain barrier in humans. After a single dose treatment with MTK mucoadhesive film, we detected MTK in the CSF of healthy subjects in a pharmacological relevant dose in the range of the IC50 (MTK IC50 < 5nM [25]). The initial results indicate that T_max_ could in fact be delayed in comparison to the plasma level results. This may indicate that MTK accumulates in the CSF faster than its clearance. This is significant as it would allow a once-a-day dosing to maintain therapeutic CSF concentrations rather than multiple times in a single day, thereby improving patient compliance. Further clinical studies are needed to determine the pharmacokinetics of CSF accumulation and clearance of MTK, as, due to its invasiveness, CSF sampling was limited to two timepoints. The overall advantages of the mucoadhesive film are the ease of administration in elderly and in patients with dysphagias. In addition, there is no liquid needed to ease swallowing. The film quickly adheres and is almost impossible to reject. Therefore, accidental swallowing of the film is highly unlikely, because the mucoadhesion, although not quantified, is strong and very quick. The film would stick somewhere in the mouth and start to dissolve before being swallowed. The full dissolution of the film is also short, estimated between 1 and 4 min depending on patient saliva amount and composition. Furthermore, this product has an improved bioavailability and portability. The one limitation of the presented film is the loading capacity, as a thin film can only be loaded up to around 60 mg of active ingredient, so dose increases in one film are limited, but could be circumvented by using, for example, one film on each side of the mouth.

MTK was approved in 1998 for the treatment of chronic asthma and is generally well tolerated. In the safety meta-analysis of the 11 multicenter, randomized, controlled studies conducted by Storms et al. (2001), a total of 3386 adult patients (aged > 15 years) and 336 pediatric patients (6 to 14 years) were enrolled. From the total adult population, 1955 patients received MTK and the remaining 1431 were on placebo or an active comparator (inhaled beclomethasone). In the meta-analysis, discontinuation as a result of adverse events (AEs) occurred in 3.7% (73 out of 1955) adult patients treated with MTK and 5.2% (61 out of 1180) receiving placebo. It was also noted that there were no notable clinical differences in the frequency of reported AEs between patients receiving MTK from those receiving placebo. The most common and frequently reported AE in the MTK and placebo groups were upper respiratory infection, asthma, and headache. In total, the incidence of patients reporting an AE was 66.5% (1300) with MTK and 71.3% (841) with placebo. In 2 of 4 extension studies in the pooled analysis, no increase in treatment discontinuation or frequency of clinical or laboratory AEs were seen in patients receiving MTK at substantially higher doses than the marketed 10 mg dose. In these two studies, patients received MTK at 200 mg for 22 weeks, 100 mg for 12 weeks and 50 mg for 28 weeks with no observed change in the safety or tolerability profile. In all four extension studies, the total continuous exposure to MTK for the adults was 698 patients for up to 6 months, 525 patients for at least 1 year, 152 patients for at least 2 years and 25 for up to 3 years, providing an overall exposure of 734 patient years [26]. In the study presented herein, six adverse events were documented occurring after taking the MTK film. Two study subjects experienced somnolence, which was possibly related to MTK. The other adverse events were classified as unlikely related and unrelated to MTK. Sleep disturbances, especially nightmares, have been reported after the use of MTK in asthma patients in children, but also in adults (reviewed in [27]), but they stop shortly after discontinuation of the treatment [28]. A retrospective study analyzing adverse reactions of MTK reported to the Netherlands Pharmacovigilance Center Lareb and the WHO Global database, VigiBase^®^ identified aggression, nightmares and suicidal ideations as the most common reported adverse drug reactions and highlighted the importance of informing patients and parents of the possibility of neuropsychiatric side effects in the clinics [29]. Of course, these neuropsychiatric side effects need to be considered in future clinical studies, although they demonstrate possible effects of MTK on the brain.

Secondly, we analyzed pharmacokinetics of MTK in a transgenic mouse model of AD, because recently it was shown that pharmacological inhibition of leukotriene signaling had beneficial effects in several mouse models of AD [30,31,32,33,34]. The leukotriene receptor antagonist MTK has been proposed as an interesting candidate for drug repurposing in AD patients [13,14,15], due to its potential to modulate neuroinflammation and improve memory in animal models of stroke [35], epilepsy [36,37] and Lewy body dementia [38]. However, the cellular and molecular mechanisms underlying MTK action on CNS remain yet poorly understood, and pre-clinical experiments are needed to shade light onto its efficacy to modulate cognitive function. Here, we evaluated the pharmacokinetics of MTK mucoadhesive film in rodents, after acute and chronic treatment following different dosing and observed dose dependent MTK serum levels after acute and long-term treatment. We observed a high variation in absorption in the acute experiment. Interestingly, MTK serum levels, 7 h post dosing, were significantly lower after a single dose of 10 mg/kg/d compared to MTK serum levels after 3 months of daily treatment with the same dose. This suggests that MTK accumulates in the body during long-term treatment, which might stabilize and/or prolong effects of MTK. However, as acute pharmacoexposure experiments were performed in C57bl6 animals and chronic pharmacoexposure experiments were performed in transgenic 5xFAD mice, we cannot discharge the hypothesis that the observed differences of MTK pharmacokinetics are related to the different genetic background of the mouse models used.

Interestingly, we could detect dose-dependent concentrations of MTK in the CSF of 5xFAD mice after chronic treatment. This constitutes an important prerequisite for future pre-clinical efficacy studies of MTK and shows the relevance of the 5xFAD mouse model as a tool to better understand the effects of MTK chronic treatment in AD. Overall, the results here presented, from both human and mice studies, pave the way for a larger clinical Phase 2 study to determine the efficacy of MTK to improve cognitive function in patients suffering from Alzheimer’s disease.

## 5. Conclusions

In summary, we have demonstrated safety, convenience and improved bioavailability of MTK in the form of the mucoadhesive film compared to the coated tablet in humans. Furthermore, we demonstrate BBB penetrance in a clinical study in human healthy subjects as well as in a preclinical mouse model for Alzheimer’s disease. This paves the way for potential therapeutic use of the oral mucoadhesive MTK film in acute and chronic neurodegenerative diseases such as Alzheimer’s disease, dementia with Lewy bodies, stroke, or spinal cord injuries.

## Figures and Tables

**Figure 1 pharmaceutics-13-00012-f001:**
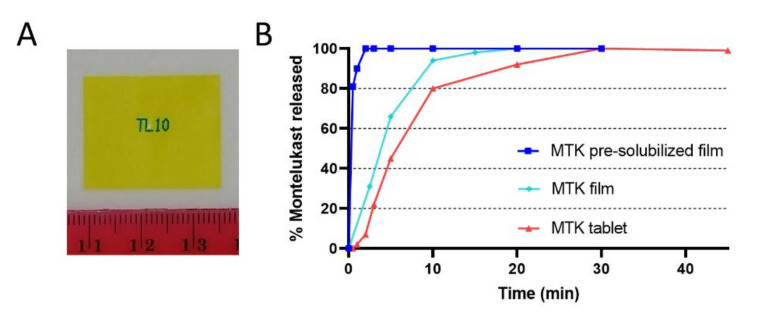
Characterization of the Montelukast oral film. (**A**) Image of a Montelukast (MTK) oral film from IntelGenx (scale in (cm)). (**B**) Graphical representation comparing the in vitro dissolution rates of (Turquoise) the MTK oral film, (Blue) pre-solubilized MTK oral film, and (Red) MTK tablet Singulair^®^. Eighty percent MTK release is reached after approximately 1 min with the pre-solubilized film, after approximately 6 min with the oral film and after 10 min with the tablet.

**Figure 2 pharmaceutics-13-00012-f002:**
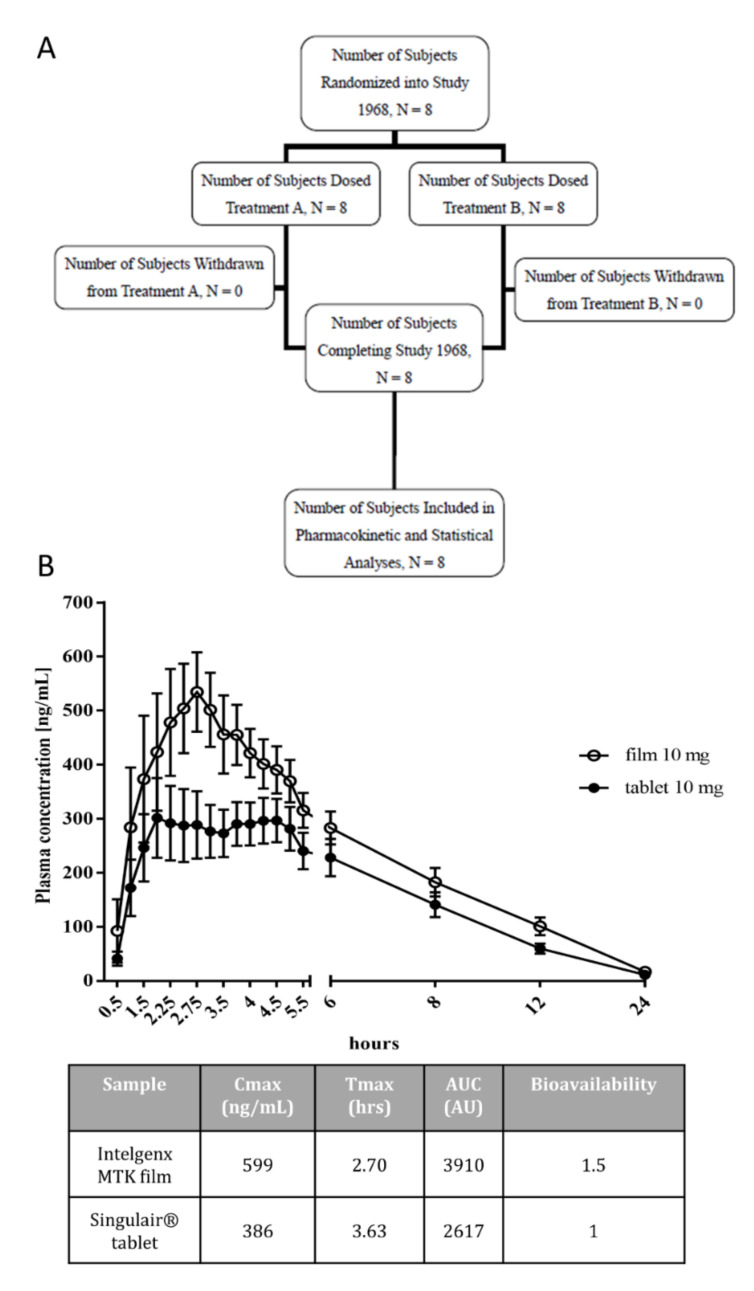
(**A**) Study plan of the clinical Phase 1 study. (**B**) Pharmacokinetic profiles of Montelukast (MTK) film and tablet. Study subjects were treated with a single dose of MTK in film or tablet form. MTK plasma concentration were then assessed and a time profile after administration of 10 mg MTK film and tablet was prepared. The 10 mg oral film had an improved bioavailability compared to the 10 mg tablet. Data are shown as mean +/− SEM.

**Figure 3 pharmaceutics-13-00012-f003:**
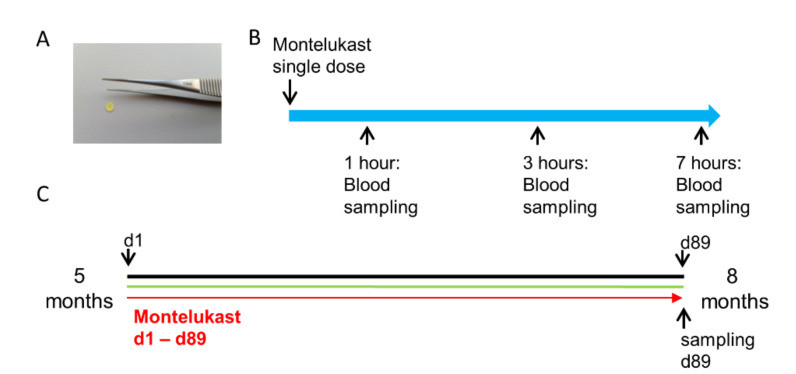
Design of the preclinical testing of MTK. Experimental setup of the pharmacoexposure and kinetics experiment in wildtype (WT) and 5xFAD mice. (**A**) Montelukast (MTK) films for mice were prepared as punches suitable for placing the films on the tongue of mice. (**B**) In the pre-experiment, C57Bl6 mice were treated with a single dose of MTK. Blood samples were taken 1, 3 and 7 h after systemic application. (**C**) Five-month-old 5xFAD transgenic mice were treated daily with vehicle or MTK in two different doses for 89 days. On the last day of treatment, mice were perfused and blood and cerebrospinal fluid (CSF) samples were collected approximately seven hours after the last application of MTK.

**Figure 4 pharmaceutics-13-00012-f004:**
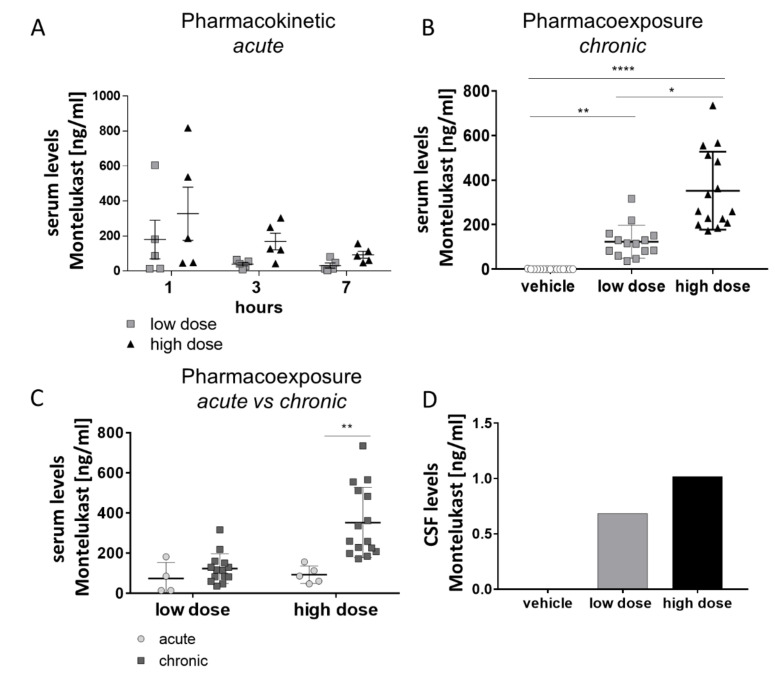
Serum levels of MTK after MTK film application. (**A**) Montelukast (MTK) concentration (high dose = 10 mg/kg/d; low dose = 3.3 mg/kg/d) in the serum decreases over 7 h. (**B**) Samples were taken approximately 7 h after last oral application of MTK by cardiac puncture. Samples were analyzed for MTK concentration by HPLC–MS/MS. There is a dose-dependent increase in serum levels of MTK. (**C**) After long-term treatment the serum concentration of MTK after 7 h is significantly higher than after single dose treatment with a dose of 10 mg/kg/d. (**D**) Pooled samples (*n* = 14–15 per group) were analyzed for MTK concentration by HPLC–MS/MS. There is a dose-dependent increase in the MTK concentration in the CSF. Data are shown as mean +/− SEM. One-way ANOVA with Bonferroni post-hoc test or Student’s t-test was performed. *p* values of *p* < 0.0001 and *p* < 0.001 were considered extremely significant (**** or ***), *p* < 0.01 very significant (**), and *p* < 0.05 significant (*).

**Figure 5 pharmaceutics-13-00012-f005:**
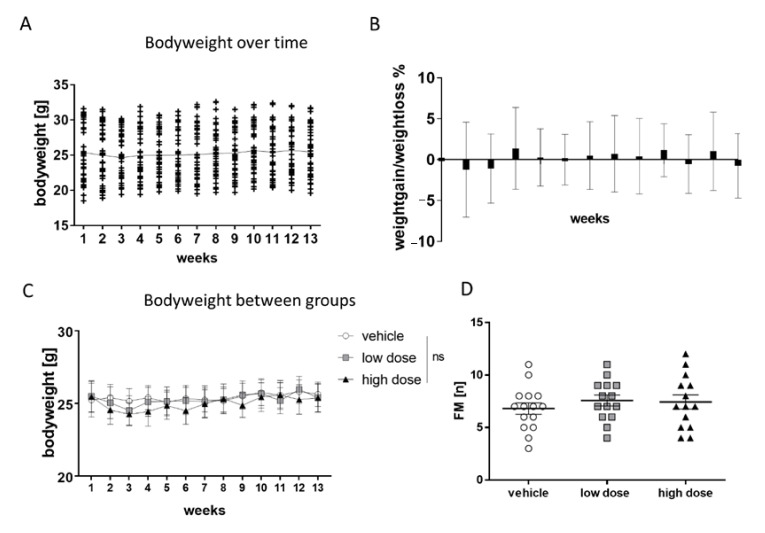
Safety profile of MTK film in mice. Body weight of the mice was assessed on a weekly basis for 13 weeks as a parameter of health. (**A**) Mean body weight did not significantly differ between the weeks. (**B**) Individual body weights did fluctuate less than +/− 10% each week. (**C**) Body weights did not significantly differ between groups (high dose = 10 mg/kg/d; low dose = 3.3 mg/kg/d). (**D**) Fecal motility did not differ significantly between all groups. Data are shown as mean +/− SEM. One-way ANOVA with Bonferroni post-hoc test or Student’s t-test was performed. *p*-values < 0.05 were considered significant.

**Table 1 pharmaceutics-13-00012-t001:** General composition of the mucoadhesive film.

Excipient	Function	Composition (%*w/w* Wet)
Water	Solvent	60.0–85.0
Mucoadhesive Polymer	Film forming polymer	5.0–15.0
Gum	Viscosity modifier	0.50–3.00
Stabilizers	Degradation prevention	0.01–0.05
Colorant/Flavor	Patient compliance	0.10–1.00
Plasticizers	Tune mechanical properties	0.10–4.00
Permeation Enhancer	Increase oral absorption	0.10–2.00
API *	Therapeutic	0–30.00

* API = active pharmacological ingredient.

**Table 2 pharmaceutics-13-00012-t002:** Mechanical characterization of the mucoadhesive film: folding endurance, elongation and tensile strength of the 10 mg Montelukast (MTK) film exposed to normal (T_0_) and freezing conditions (T 15 days and 30 days). Results are shown as the average of three to eighth films (mean with SD).

Time	T_0_	15 Days	30 Days
Folding endurance	>10	>10	>10
Elongation at break (%)	62 ± 6	69 ± 7	65 ± 9
Tensile strength (kPa)	128 ± 10	115 ± 15	123 ± 13

**Table 3 pharmaceutics-13-00012-t003:** Study demographics of the clinical Phase 1 study.

Parameter	Age	Body Mass Index (BMI)
**Mean**	44 ± 6	26 ± 3
**Median**	46	26.5
**Range**	31–50	22.4–29.4

**Table 4 pharmaceutics-13-00012-t004:** Comparative bioavailability analysis for plasma MTK buccal film versus Singulair^®^ tablet in humans.

Parameter	Geometric Means		Ratio of Geometric Means (Test/Reference in (%))	90% Confidence Interval	Intra-Subject CV (%)
	Film	Singulair^®^			
AUCt (ng×h/mL)	3673	2409	152.46	101.02–230.10	45.58
AUCinf (ng×h/mL)	3827	2499	153.15	99.77–235.1	45.88
C_max_ (ng/mL)	554	338	163.89	99.12–270.99	57.05
T_max_ (h)	2.63	3.63			

CV: Coefficient of Variation. AUCt: area under the concentration–time curve from time zero until the last measurable concentration or last sampling time t, whichever occurs first. AUCt is estimated using the trapezoidal method. AUCt/AUCinf: the proportion of AUCinf covered by the actual sample schedule. AUCinf: area under the concentration–time curve from time zero to infinity, calculated as AUCt + Clast/λ, where Clast is the last measurable concentration. C_max_: the maximal observed plasma concentration. T_max_: time when the maximal plasma concentration is observed.

**Table 5 pharmaceutics-13-00012-t005:** Descriptive statistics for MTK concentrations in cerebrospinal fluid at 3.0 and 7.0 h post-dose after a single dose of 10 mg MTK oral mucoadhesive film in humans.

Descriptive Statistics	3.0 h	7.0 h
N	8	8
Max (ng/mL)	4.2	4.7
Min (ng/mL)	3.2	3.8
Median (ng/mL)	3.4	4.2
Mean (ng/mL)	3.6	4.2
Std Dev (ng/mL)	0.36	0.31
CV (%)	10.2	7.19

**Table 6 pharmaceutics-13-00012-t006:** Adverse events reported during the study.

System Organ Classification/ Preferred Term (PT)	Reported Incidence by Treatment Group	
	MTK Film (N = 8)	MTK Tablet (N = 8)
**Nervous System Disorders**		
Somnolence	2 (25.0%))	0 (0.0%)
Headache	1 (12.5%)	0 (0.0%)
**Skin and Subcutaneous Tissue Disorders**		
Pruritus	1 (12.5%)	0 (0.0%)
Mechanical urticaria	1 (12.5%)	0 (0.0%)
**Musculoskeletal and Connective Tissue Disorders**		
Back pain	1 (12.5%)	0 (0.0%)
**Total**	6 (75.0%)	0 (0.0%)

MTK Film: Montelukast 10 mg Oral Film; Lot No: 16E-MOT010-0001, by IntelGenx Corp. MTK Tablet: Singulair^®^ 10 mg Tablets; Lot No: L044452, by Merck Sharp & Dohme Ltd., Hertfordshire, UK.

**Table 7 pharmaceutics-13-00012-t007:** Severity, relationship to study and action taken for adverse events.

Treatment Group	Severity			Relationship to Drug				Intervention		
	Mild	Moderate	Severe	Unrelated	Unlikely	Possible	Probable	Pharmacologic	Other	None
A	6	0	0	3	1	2	0	2 *	1 *	4
B	0	0	0	0	0	0	0	0	0	0
Total	6	0	0	3	1	2	0	2	1	4

* One subject was treated with both pharmacological and other interventions for adverse effect (AE) back pain in Period 1.

## Data Availability

Authors declare availability of data and material upon request, as far as possible within ethical and privacy constraints.

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
