# Peer review of "Improved Bioavailability of Montelukast through a Novel Oral Mucoadhesive Film in Humans and Mice"

_pharmaceutics, 2020, doi:10.3390/pharmaceutics13010012_

Round 1
Reviewer 1 Report
Manuscript Pharmaceutics-1018983 is reporting the development of a mucoadhesive film containing montelukast. The author proved that the in vitro dissolution of the mucoadhesive formula is much better than that of the tablet. In human clinical study they proved that the bioavailability of the new formula is better than the recently applied and registered montelukast tablet. Beside the plasma levels, they measured the CSF levels of the drug after the mucoadhesive administration and they found that the montelukast level was well detectable. In special mice that are the model of Alzheimer disease they also proved that the mucoadhesive formula can reach remarkable levels of montelukast both in plasma and CSF. The authors conclude that their new formula can be beneficial both for asthmatic and neurodegenerative patients.
The followings must be answered or modified before the accepting the manuscript for publication
- Line 36-43: The criteria of the abstract must be deleted
- Line 66-67: each tablet is inadequate in case of dysphagia or artificial ventilation; therefore this statement must be modified as a general one and not specified to montelukast tablet
- Line 81: Although there are some information about the link between Alzheimer disease and montelukast in the Result and the Discussion, however, few sentences must be written into the Introduction about this issue.
- Line 266: Table 3 is useless, because does not provide any additional information as compared with Figure 1 with its legend. Thus, Table 3 must be deleted.
- Line 277: The authors claim that the in vitro dissolution study demonstrates the montelukast release in the stomach. This statement must be modified, because the pH of the buffer was 6.8 (Line 94) that is far from the acidity of the stomach.
- Line 284: Table 4 can be simplified, the BMI values would be enough to present, hence the height and the weight gives the BMI.
- Line 286: Figure 2 table – it is not clear how the Tmax for Singulair can be 3.6 h, if the peak concentration of montelukast has reached around 2.0 h (Figure 2B)? Between 2-4.5 h no changes can be read from the plasma curve.
- Line 299-300. A short explanation (1-2 sentences) about the importance of the geometric mean calculation in bioavailability analysis would be helpful for those readers who are not familiar with these calculations.
- Line 321: Have the CSF levels of montelukast been investigated after the administration of Singulair tablet? No data presented in the manuscript. If not, what was the reason? Without such a control it is hard to evaluate the significance of the CSF penetration of montelukast from the mucoadhesive formula. The same is true for the mice study (Line 377)
- Line 377: Figure 4A – The serum levels after 1 h of the administration is very diverse among the 5 animals of each group. It suggests that the rate of absorption was individually very different. This must be addressed in the Result and the Discussion.
- Line 397: The details of fecal boli counts must be included into the Methods along together with the schedule of the body weight measurement
- Line 498: Although COVID-19 currently is a magical keyword, this work has no any evidential benefit for COVID-19 disease or complications, therefore I recommend to delete this speculation (this is also true for the mentioning it in the abstract).
Author Response
Reviewer 1
- Line 36-43: The criteria of the abstract must be deleted
Response:
We are sorry for this mistake, we missed this when formatting the manuscript. We thank Reviewer 1 for pointing that out. Criteria of the abstract were now deleted in the revised version.
- Line 66-67: each tablet is inadequate in case of dysphagia or artificial ventilation; therefore this statement must be modified as a general one and not specified to montelukast tablet
Response:
Thank you very much for pointing this out. We now changed the sentence to:
„Besides the physico-chemical and genetic basis for the insufficient uptake and bioavailability of MTK in its current tablet form, a further drawback is the general inadequateness of the tablets for patients suffering from dysphagia such as elderly patient, for patients with dementias, and for patients that require intubation or ventilation.”
- Line 81: Although there are some information about the link between Alzheimer disease and montelukast in the Result and the Discussion, however, few sentences must be written into the Introduction about this issue.
Response: We added a short paragraph in the introduction:
„There is already a certain rational for MTK to be repurposed for Alzheimer’s Disease (AD). Leukotriene production is elevated in the AD brain, leukotrienes are involved in various aspects of AD pathologies, in particular neuroinflammation and neuronal cell death, and genetic and pharmacologic blockage of leukotrienes in animal models of neurodegenerative diseases including AD has demonstrated efficacy in reducing the pathology an promoting cognitive functions. Finally, MTK is associated with a reduced risk of dementias, and a recent case report suggested that MTK might improve cognitive function in patients with dementias.“
- Line 266: Table 3 is useless, because does not provide any additional information as compared with Figure 1 with its legend. Thus, Table 3 must be deleted.
Response:We originally added the table to offer the possibility of a simplified presentation. We are fine with deleting the table, and did that in the revised version.
- Line 277: The authors claim that the in vitro dissolution study demonstrates the montelukast release in the stomach. This statement must be modified, because the pH of the buffer was 6.8 (Line 94) that is far from the acidity of the stomach.
Response: We entirely agree with this argument. We corrected this and modified the statement as the reviewer suggested.
- Line 284: Table 4 can be simplified, the BMI values would be enough to present, hence the height and the weight gives the BMI.
Response: We simplified the table by deleting the values for height and weight, as the reviewer says BMI is enough.
- Line 286: Figure 2 table – it is not clear how the Tmax for Singulair can be 3.6 h, if the peak concentration of montelukast has reached around 2.0 h (Figure 2B)? Between 2-4.5 h no changes can be read from the plasma curve.
Response: Thanks for the comment. For explanation: The Tmax is based on the time at which the maximum concentration is achieved for each individual and not based on the average obtained from the graphical representation. Singulair demonstrates a plateau-like maximum concentration where the Tmax will vary from one subject to the other. The average representation of the plasma levels is something misleading for the assessment of Tmax looking at individual data is required to assess the Tmax.
Therefore, we now added the following information in the text: „Tmax was calculated based on the time at which the maximum concentration is achieved for each individual and not based on the average obtained from the graphical representation. Singulair demonstrated a plateau-like maximum concentration where Tmax varied from one subject to the other. The inter-individual variation of Tmax was less in the MTK mucoadhesive resulting in a peak at 2.7 hrs.”
- Line 299-300. A short explanation (1-2 sentences) about the importance of the geometric mean calculation in bioavailability analysis would be helpful for those readers who are not familiar with these calculations.
Response: Thanks for this comment. We added a short explanation why we used the geometric mean for our data: “In this study the parameters of AUC and Cmax don’t follow a normal distribution, but a lognormal distribution. Therefore, as favored the use of the geometric mean over the arithmetic mean.”
- Line 321: Have the CSF levels of montelukast been investigated after the administration of Singulair tablet? No data presented in the manuscript. If not, what was the reason? Without such a control it is hard to evaluate the significance of the CSF penetration of montelukast from the mucoadhesive formula. The same is true for the mice study (Line 377)
Response: We were unfortunately not able to investigate the potential of MTK to cross the blood-brain-barrier after using the singulair® tablet. As CSF sampling is rather invasive, we were limited to the analysis of MTK in the CSF only after using the newly developed MTK-film. Also, we had demonstrated already in our publication Marschallinger et al., 2015, Nat. Commun., that MTK is detected in the CSF after oral intake of MTK tablets at a pharmacological relevant level. In the present work, we aimed to demonstrate that MTK (after application of the MTK film) is generally able to cross the blood-brain-barrier. As stated above, we did not have the chance to compare CSF levels achieved by the two different formulations. As a further point, the ability of MTK to cross the blood-brain-barrier is of importance when it comes to its repurposing for neurodegenerative diseases. That is why we used the mucoadhesive film in the here presented mouse study.
- Line 377: Figure 4A – The serum levels after 1 h of the administration is very diverse among the 5 animals of each group. It suggests that the rate of absorption was individually very different. This must be addressed in the Result and the Discussion.
Response: Thank you vey much for pointing this out. We now addressed the fact that the individual absorption varies in the results and the discussion part. For example, „The animals receiving the high dose treatment showed, in general, higher MTK serum concentrations, although we observed a high inter-individual variation in serum concentrations (Fig 4A). This suggests variation in the rate of absorption between individuals.”
- Line 397: The details of fecal boli counts must be included into the Methods along together with the schedule of the body weight measurement
Response: Thanks for pointing this out. Details about fecal motility and bodyweight were added to the method section.
- Line 498: Although COVID-19 currently is a magical keyword, this work has no any evidential benefit for COVID-19 disease or complications, therefore I recommend to delete this speculation (this is also true for the mentioning it in the abstract).
Response: We agree with the reviewer, and we took out all statements regarding COVID-19.
Reviewer 2 Report
The manuscript entitled “Improved Bioavailability of Montelukast Through a Novel Oral Mucoadhesive Film in Humans and Mice” by Johanna Michael developed a novel MTK oral mucoadhesive film to circumvent the limitations of MTK in its tablet form.
Generally speaking, this work presents some findings but there were still a few issues to be settled and the manuscript needs to be improved in some aspects before its acceptance. My recommendation is that the manuscript needs a minor revision in its current. Below are few of questions and recommendation for the authors.
- If used in the future clinical application, what are the advantages of the mucoadhesive film, and what is the limitation of the mucoadhesive film?
- The adhesive strength of the mucoadhesive film should be provided.
- How about the degradation behavior of the mucoadhesive film?
- What is the biocompatibility of the mucoadhesive film, for patients who swallow the mucoadhesive film by mistake, can the mucoadhesive film eventually be excreted?
- In fact, from the perspective of material preparation in the manuscript, it seems possible to develop other drug delivery materials based on the mucoadhesive film. Therefore, please explain whether similar studies about the mucoadhesive film have been reported? This is essential to illustrate the innovation of materials.
Author Response
Reviewer 2
- If used in the future clinical application, what are the advantages of the mucoadhesive film, and what is the limitation of the mucoadhesive film?
Response: We added the following part in the discussion: The overall advantages of the mucoadhesive film are the ease of administration in elderly and in patients with dysphagias..Also, there is no liquid needed that would ease swallowing. The film quickly adheres and is almost impossible to reject. Furthermore, this product has an improved bioavailability and portability. The one limitation of the presented film is the loading capacity, as a thin film can only be load up to around 60mg of active ingredient, so dose increases in one film are limited, but could be circumvented by using for example one film on each side of the mouth.
- The adhesive strength of the mucoadhesive film should be provided.
Response: We do agree that it would be a valuable further piece of information. However, at the moment we do not have these sort of data available yet. We are planing in future experiments do adress this question.
- How about the degradation behavior of the mucoadhesive film?
Response: Actually, we do have some data regarding stability. The 10 mg filmd has completed a stability study of 2 years under ICH long term stability condition (25°C/60%RH) and 6 months under ICH accelerated conditions (40°C/75%RH) without any impurities out of specifications. We added this infrmration in the result section.
- What is the biocompatibility of the mucoadhesive film, for patients who swallow the mucoadhesive film by mistake, can the mucoadhesive film eventually be excreted?
Response: This is, of course, an interesting question. The mucoadhesion, although not quantified, is strong and very quick and swallowing the film is almost impossible as it will stick somewhere in the mouth and will start to dissolve before going down. The full dissolution of the film is also short, estimated between 1 and 4 minutes depending on patient saliva amount and composition.
- In fact, from the perspective of material preparation in the manuscript, it seems possible to develop other drug delivery materials based on the mucoadhesive film. Therefore, please explain whether similar studies about the mucoadhesive film have been reported? This is essential to illustrate the innovation of materials.
Response: We thank Reviewer 2 for this comment and added examples of similar drug delivery systems that are already on the US market.„In the US there some other commercial products that also use a film as drug delivery system. For example, within the course of treating opioid addiction a sublingual film of Suboxone®, containing buprenorphine and naloxone, can be used. Other examples would be Belbuca®, a buccal film containing buprenorphine used for pain treatment and Kynmobi, apomorphine hydrochloride, a sublingual treatment used in parkinsons disease. However, the number of products on the market is very limited, and several other products are under development.